# Long-Term Efficacy and Safety of Rifaximin in Japanese Patients with Hepatic Encephalopathy: A Multicenter Retrospective Study

**DOI:** 10.3390/jcm11061571

**Published:** 2022-03-12

**Authors:** Hideto Kawaratani, Yasuteru Kondo, Ryoji Tatsumi, Naoto Kawabe, Norikazu Tanabe, Akira Sakamaki, Kazuo Okumoto, Yoshihito Uchida, Kei Endo, Takumi Kawaguchi, Tsunekazu Oikawa, Yoji Ishizu, Shuhei Hige, Taro Takami, Shuji Terai, Yoshiyuki Ueno, Satoshi Mochida, Yasuhiro Takikawa, Takuji Torimura, Tomokazu Matsuura, Masatoshi Ishigami, Kazuhiko Koike, Hitoshi Yoshiji

**Affiliations:** 1Department of Gastroenterology, Nara Medical University, Kashihara 634-8521, Japan; yoshijih@naramed-u.ac.jp; 2Department of Hepatology, Sendai Kousei Hospital, Sendai 980-0873, Japan; yasuteru@ebony.plala.or.jp; 3Department of Gastroenterology, Sapporo Kosei General Hospital, Sapporo 060-0033, Japan; nyantyu_seijin@yahoo.co.jp (R.T.); shuhei.hige@ja-hokkaidoukouseiren.or.jp (S.H.); 4Department of Gastroenterology and Hepatology, Fujita Health University School of Medicine, Aichi 470-1192, Japan; kawabe@fujita-hu.ac.jp; 5Department of Gastroenterology and Hepatology, Graduate School of Medicine, Yamaguchi University, Ube 755-8505, Japan; norikazu@yamaguchi-u.ac.jp (N.T.); t-takami@yamaguchi-u.ac.jp (T.T.); 6Division of Gastroenterology and Hepatology, Graduate School of Medical and Dental Sciences, Niigata University, Niigata 951-8510, Japan; saka-a@med.niigata-u.ac.jp (A.S.); terais@med.niigata-u.ac.jp (S.T.); 7Department of Gastroenterology, Yamagata University Faculty of Medicine, Yamagata 990-9585, Japan; okumoto@med.id.yamagata-u.ac.jp (K.O.); y-ueno@med.id.yamagata-u.ac.jp (Y.U.); 8Department of Gastroenterology and Hepatology, Faculty of Medicine, Saitama Medical University, Saitama 350-0495, Japan; uchida.yoshihito@1972.saitama-med.ac.jp (Y.U.); smochida@saitama-med.ac.jp (S.M.); 9Division of Hepatology, Department of Internal Medicine, Iwate Medical University, Morioka 028-3694, Japan; keiendo@iwate-med.ac.jp (K.E.); ytakikaw@iwate-med.ac.jp (Y.T.); 10Division of Gastroenterology, Department of Medicine, Kurume University School of Medicine, Kurume 830-0011, Japan; takumi@med.kurume-u.ac.jp (T.K.); tori@med.kurume-u.ac.jp (T.T.); 11Division of Gastroenterology and Hepatology, Department of Internal Medicine, The Jikei University School of Medicine, Tokyo 105-8461, Japan; oitsune@jikei.ac.jp (T.O.); matsuurat@jikei.ac.jp (T.M.); 12Department of Gastroenterology and Hepatology, Nagoya University Graduate School of Medicine, Aichi 466-8550, Japan; y-ishizu@med.nagoya-u.ac.jp (Y.I.); masaishi@med.nagoya-u.ac.jp (M.I.); 13Kanto Central Hospital, Tokyo 158-8531, Japan; kkoike.tky@gmail.com

**Keywords:** cirrhosis, hepatic encephalopathy, Japanese, long-term, rifaximin

## Abstract

Background: Rifaximin is commonly used for hepatic encephalopathy (HE). However, the effects of long-term treatment for Japanese people are limited. Therefore, this study aimed to investigate the effects and safety of long-term treatment with rifaximin on HE. Methods: A total of 215 patients with cirrhosis administered with rifaximin developed overt or covert HE, which was diagnosed by an attending physician for >12 months. Laboratory data were extracted at pretreatment and 3, 6, and 12 months after rifaximin administration. The long-term effect of rifaximin was evaluated, and the incidence of overt HE during 12 months and adverse events was extracted. Results: Ammonia levels were significantly improved after 3 months of rifaximin administration and were continued until 12 months. There were no serious adverse events after rifaximin administration. The number of overt HE incidents was 9, 14, and 27 patients within 3, 6, and 12 months, respectively. Liver enzymes, renal function, and electrolytes did not change after rifaximin administration. Prothrombin activity is a significant risk factor for the occurrence of overt HE. The serum albumin, prothrombin activity, and albumin–bilirubin (ALBI) scores were statistically improved after 3 and 6 months of rifaximin administration. Moreover, the same results were obtained in patients with Child–Pugh C. Conclusions: The long-term rifaximin treatment was effective and safe for patients with HE, including Child–Pugh C.

## 1. Introduction

Hepatic encephalopathy (HE) is a common complication of patients with liver cirrhosis, causing neuropsychological and neuromuscular dysfunctions in various degrees. HE is closely associated with a poor prognosis and reduced health-related quality of life [1,2]. HE is defined as a brain dysfunction caused by liver insufficiency and/or portosystemic shunting (PSS), manifesting as a wide spectrum of neurological or psychiatric abnormalities ranging from subclinical alterations to coma [3]. HE is classified into two types: covert and overt. Covert HE is involved in minimal HE and a Grade I coma; it is presented as a slight abnormality, and cannot be diagnosed, except by a psychometric or neuropsychological examination or clinical findings usually not reproducible, whereas overt HE is involved in Grade II to IV comas and presents obvious consciousness and motor activity disorders [4].

The prognosis of liver cirrhosis reportedly worsens when overt HE occurs, with a mortality rate within 1 year of 64% and within 5 years of 85%. Covert HE occurs in 80% of patients with liver cirrhosis, and covert HE, as well as overt HE, can impair the prognosis and increase the risk of hospitalization [5].

In the Japanese guideline, synthesized disaccharides are the first-line treatment for HE, which decreases the production and absorption of ammonia in the gut [6]. One of the second-line treatments for HE is poorly absorbed antibiotics. In Japan, poorly absorbed antibiotics, such as kanamycin or polymyxin B, had been used for HE; however, those were not approved by the health insurance. Rifaximin is an oral non-absorbable antibiotic that decreases ammonia-producing enteric bacteria [7]. Rifaximin treatment significantly decreases ammonia levels, prevents HE recurrence, reduces the risk of hospitalization involving HE, or prolongs overall survival [8,9]. The data of long-term rifaximin treatment have been reported from Europe and the USA [10,11]. Rifaximin (Rifxima^®^; ASKA Pharmaceutical, Tokyo, Japan) was approved for the HE treatment in November 2016 in Japan. However, only data from clinical trials were available up to 12 weeks of administration [12], and data on the long-term efficacy and safety of rifaximin treatment have been limited in Japan [13,14,15,16]. Moreover, all data were obtained from a single-center retrospective study. As the gastrointestinal microbiome differs by race [17,18,19], the efficacy of rifaximin may differ between Japanese and Europeans or Americans. Therefore, the efficacy of rifaximin in Japanese people should be investigated. Furthermore, the usefulness of rifaximin in patients with Child–Pugh C was not sufficiently evaluated. Thus, in this multicenter retrospective study, we investigated the usefulness and safety of long-term rifaximin treatment.

## 2. Materials and Methods

### 2.1. Patients

This was a multicenter retrospective observational cohort study of 215 consecutive patients with liver cirrhosis who were continuously administered 1200 mg t.i.d of rifaximin for >12 months between January 2017 and July 2020. Rifaximin was administered to patients with over Grade I coma. Patients undergoing hemodialysis were administered warfarin, and with a total bilirubin of >5.0 mg/dL were excluded from this study. The primary outcome was the effectiveness of long-term rifaximin treatment, and secondary outcomes were the evaluation of the safety of long-term rifaximin treatment.

### 2.2. Ethics

This study was conducted in accordance with the ethical guidelines of the 1975 Declaration of Helsinki and approved by the Ethical Review Committees of each participating institution for this study. Informed consent was obtained from all participants.

### 2.3. Data Collection

Necessary data were retrospectively extracted from each participating institution before and 3, 6, and 12 months after administering rifaximin. Age; sex; height; body weight; body mass index; hemoglobin level (Hb); platelet count (Plt); prothrombin (PT) activity; serum levels of aspartate aminotransferase (AST), alanine aminotransferase (ALT), albumin (Alb), total bilirubin (T-Bil), blood urea nitrogen (BUN), creatinine (Cr), estimated glomerular filtration (eGFR), ammonia (NH_3_), sodium (Na), and potassium (K); and medical history of HE, spontaneous bacterial peritonitis (SBP), coexistence of hepatocellular carcinoma (HCC), stage of HCC, and coexistence of PSS were extracted. PSS was evaluated with enhanced CT or MRI, and a maximum vessel diameter of 5 mm indicated the existence of PSS. Moreover, the albumin–bilirubin (ALBI) score and model for end-stage liver disease (MELD) score were calculated. The occurrence or recurrence of HE was defined as the occurrence of HE Grade II or higher.

### 2.4. Statistical Analysis

Data are presented as mean ± standard deviation. To compare continuous variables at baseline and after rifaximin administration at 3, 6, 12 months, repeated measurement analysis of variance was used. A Kaplan–Meier curve was created with Gray’s test for evaluating cumulative HE occurrence. A Cox proportional hazard model was used to determine the factors associated with HE occurrence in univariate and multivariate analyses. Parameters with a *p*-value of <0.1 in univariate analysis were entered into multivariate analysis. Statistical comparison was performed using the SPSS version 24.0 software (IBM Corp., Tokyo, Japan), and a *p*-value of <0.05 was considered statistically significant.

## 3. Results

### 3.1. Patient Characteristics

The patient characteristics are shown in Table 1. The mean age of 215 patients receiving rifaximin for >12 months was 69.7 (range, 23–94) years, with 64.1% comprising men (138 men and 77 women). The etiology of cirrhosis was hepatitis B (22), hepatitis C (37), alcohol-induced (66), nonalcoholic steatohepatitis (29), and other causes, including autoimmune hepatitis of primary biliary cholangitis (61). Among them, 163 patients (75.8%) had a history of HE. A total of 14, 174, and 27 patients had Child–Pugh classes A, B, and C, respectively. In 116 (54.0%) patients, rifaximin was added to lactulose. The complications of cirrhosis were associated with esophageal varices (119 patients), PSS (107 patients), splenomegaly (160 patients), ascites (79 patients), and HCC (74 patients). Stage 1, 2, 3, and 4 HCCs were noted in 27, 30, 11, and 6 patients, respectively. No patients had complicated SBP, and the laboratory examinations of pretreatment are shown in Table 2.

### 3.2. Efficacy of Long-Term Rifaximin Use

Ammonia levels were significantly decreased by administering rifaximin after 3 months, which was continued for 12 months (Figure 1A). Overt HE occurred in 9, 14, and 27 patients within 3, 6, and 12 months, respectively, and 3, 6, 12 patients were hospitalized because of HE within 3, 6, and 12 months. The cumulative incidence of the occurrence of HE is shown in Figure 2. The cumulative incidence of the occurrence of HE within 12 months was 0.126 (27 times in 215 cases).

### 3.3. Safety of Long-Term Rifaximin Use

Hb, Plt, AST, and ALT were not affected at 3, 6, and 12 months of rifaximin treatment. BUN, Cr, and eGFR were also not affected at 3, 6, and 12 months of rifaximin treatment (Figure 3A–C). The serum Na and K levels also did not differ at 3, 6, and 12 months of rifaximin treatment (Figure 3D,E). Moreover, no patient experienced serious adverse events (AEs) related to renal function. T-bil was not affected at 3, 6, and 12 months of rifaximin treatment (Figure 4A). Conversely, the PT activity increased after 3, 6, and 12 months of rifaximin treatment (Figure 4B), the serum albumin level increased after 3 and 6 months of rifaximin treatment (Figure 4C), and the ALBI score decreased after 3 and 6 months of rifaximin treatment (Figure 4D). However, the MELD score did not differ after rifaximin treatment (Figure 4E).

No serious AEs, such as liver failure or ruptured esophagogastric varices, occurred for 12 months. A total of 12 patients experienced diarrhea, which was improved by the short-term rifaximin reduction or administration of probiotics. No patient had SBP during the rifaximin treatment.

### 3.4. The Efficacy and Safety of Long-Term Rifaximin Use in Patients with Child–Pugh C

We evaluated the data limited to Child–Pugh C. There were 27 patients of Child–Pugh C. Ammonia levels were significantly decreased by administering rifaximin after 3 months and were continued for 12 months in patients of Child–Pugh C (Figure 1B). AST, ALT, BUN, Cr, and eGFR were not affected at 3, 6, and 12 months of rifaximin treatment. The serum Na and K levels also did not differ at 3, 6, and 12 months of rifaximin treatment (Na: 136.7 ± 4.4, 137.5 ± 3.2, 137.5 ± 4.0, 136.6 ± 3.5, respectively; K: 4.1 ± 0.5, 4.3 ± 0.7, 4.1 ± 0.6, 4.1 ± 0.6, respectively). T-bil was not affected at 3, 6, and 12 months of rifaximin treatment. The PT activity and serum albumin level increased after 3 months of rifaximin treatment (Figure 5A,B). Furthermore, the ALBI score improved after 3 months of rifaximin treatment (Figure 5C). However, the MELD score did not differ at 3, 6, and 12 months of rifaximin treatment (Figure 5D).

### 3.5. Risk Factors of HE Occurrence during Rifaximin Treatment

In the Cox regression analysis, univariate analysis identified sex (male), presence of PSS, presence of HCC, Alb level at baseline, and PT activity at baseline as potential risk factors influencing HE occurrence. Multivariate Cox analysis revealed that only PT activity at baseline was significantly associated with HE occurrence (HR, 0.98; 95% CI, 0.95–1.00; *p* = 0.049) (Table 3).

## 4. Discussion

In this multicenter retrospective cohort study, the efficacy and safety of long-term rifaximin use have been clarified. The serum ammonia level significantly decreased 3 months after initiating treatment as compared to the pretreatment level, and the effect was maintained until 12 months. Moreover, in patients with Child–Pugh C, the serum ammonia level also significantly decreased 3 months after treatment, and the effect was maintained until 12 months. Furthermore, the limited occurrence of overt HE was reported (27 patients, 12.6%) at 12 months, suggesting that long-term rifaximin treatment has good efficacy and can be used continuously for a long time without discontinuation. This study showed that low PT activity is a significant predictor of HE occurrence. Although the West Haven criteria are mainly used to assess HE [20], they are insufficient for identifying subclinical encephalopathy. Ammonia levels alone do not add any diagnostic, staging, or prognostic value in HE patients with chronic liver disease [21]. However, using ammonia-lowering drugs, repeated ammonia measurements are helpful to test the efficacy. The Trail Making Test or Stroop test is useful to detect covert encephalopathy [22,23]. In this retrospective study, the Trail Making Test or Stroop test was not performed. Thus, the effects of rifaximin on covert HE or hyperammonemia were determined based on ammonia levels or the attending physician’s judgment of the clinical symptoms.

Tyrosine kinase inhibitors or immune checkpoint inhibitors are usually used for advanced HCC or HCC that is difficult or refractory to local treatment in patients with Child–Pugh A. A Phase I clinical trial was conducted on the use of Lenvatinib for treating advanced HCC, Grade 3, and HE occurred in 10% of the participants [24]. Presumably, high doses of Lenvatinib induce HE, and several reports demonstrated HE after Lenvatinib administration [25,26]. However, the mechanism for the increased ammonia has not been clarified [27]. On the contrary, in the REFLECT study, in a phase III clinical trial to compare the efficacy and safety of Lenvatinib versus sorafenib for the first-line treatment of advanced HCC, Lenvatinib was non-inferior to sorafenib in terms of overall survival. Although this study reported serious adverse effects caused by Lenvatinib, HE was not included [28]. In our study, Lenvatinib was administered in only three patients, and it did not influence the occurrence of HE.

Rifaximin is used for Clostridioides difficile infections (CDI) and HE. Rifaximin was first approved in Italy in 1985 [29]. However, it was not approved until November 2016 in Japan. Rifaximin prevents the occurrence and recurrence of overt HE and decreases the HE rates during hospitalization [9]. The safety and effectiveness of long-term treatment with rifaximin have already been reported in Europe or the USA [10,11]. Conversely, in Japan, its safety and effectiveness were evident after 12 weeks of treatment only [12], and little is known about the efficacy of long-term treatment with rifaximin in Japan [13,14,15,16]. Moreover, little evidence of rifaximin treatment has been reported for patients with Child–Pugh C. A meta-analysis showed that the efficacy of lactulose and rifaximin are comparable in improving HE [8,30]. The most-reported AEs with lactulose were diarrhea, distaste to lactulose, and abdominal bloating, which improved the following reduction of lactulose dosing [31]. Conversely, AEs with rifaximin are diarrhea and constipation, and severe CDI sometimes occurs. The CDI rate remained stable with long-term rifaximin treatment [11]. Direct head-to-head evidence of the long-term safety/tolerability of rifaximin versus lactulose is limited to a single-center, retrospective study [32], and percentages of AEs (diarrhea, flatulence, and abdominal pain) were significantly higher during the lactulose than during the rifaximin treatment. A randomized, double-blind controlled trial that compared the results of rifaximin plus lactulose therapy with lactulose therapy alone showed that the number of patients who experienced a complete recovery from HE was significantly higher in the rifaximin plus lactulose group than that in the lactulose single group. Moreover, rifaximin plus lactulose decreased the mortality rate and death from sepsis and shortened hospitalization [33]. A previous Japanese phase III study showed that AEs occurred in 13.4% of patients, and gastrointestinal AEs occurred in 6.4% of patients at 12 weeks [12]. Regarding AEs in this study, 12 (5.6%) patients had diarrhea, which improved by decreasing the dose of rifaximin or administering probiotics, and rifaximin therapy was not discontinued due to AEs. As this study excluded patients who discontinued rifaximin before 12 months for various reasons (including AEs), the exact incidence of AEs due to rifaximin is not known. Mild complications may not be extracted by physicians that are not part of a clinical phase III study.

In our study, hepatic reserve such as Alb, PT activity, and ALBI score improved 3 months after rifaximin treatment. The ALBI score, which uses only Alb and T-Bil, is a useful tool for evaluating liver function [34,35]. Studies have reported that the ALBI score can be used instead of the Child–Pugh score [36]. Moreover, the MELD score, which uses T-Bil, Cr, and PT-INR, can also predict the survival rate of patients with liver cirrhosis caused by infection, variceal bleeding, fulminant liver failure, and alcoholic hepatitis [37]. In this multicenter study, Child–Pugh was not calculated because the number of ascites was not evaluated. Instead, the ALBI score was easily evaluated, and the score statistically improved after 3 months of rifaximin administration, including Child C. The MELD score did not differ by administering rifaximin; however, the average MELD score decreased after 3 months of rifaximin administration. The MELD score consists of three items, but only one item (PT activity) was significantly changed in this study. As coagulation factors are produced by the liver, the PT is prolonged because coagulation factors are decreased due to cirrhosis or hepatitis. Treatment with a nucleoside analog for hepatitis B [38], direct-acting antivirals for hepatitis C [39], or balloon occulted retrograde transfemoral obliteration for gastric varices [40] improved Alb and PT activity. This study also showed that Alb and PT activity improved after administering rifaximin, which suggests that long-term rifaximin treatment improves hepatic reserve. Several possible reasons should be considered for the improvement of the hepatic reserve by administering rifaximin. First, rifaximin treatment improved HE, which also improved QOL and nutritional status. A recent report showed that long-term rifaximin treatment decreased blood ammonia levels and sustained low ammonia levels, resulting in an improvement in the nutritional status as assessed by the CONUT score [13]. The curative effect of rifaximin on improving dietary intake may be partly due to the improvement of HE. Second, nutrient absorption is improved by improving small intestinal bacterial overgrowth (SIBO). Since SIBO is a factor in malabsorption from the digestive tract, improving these factors may increase nutrient absorption. Patients with HE have a high prevalence of SIBO and frequently had cirrhosis as a result of impaired intestinal motility and delayed bowel transit time, including malabsorption [41]. As rifaximin is a poorly absorbed oral antibiotic, SIBO was improved by administering rifaximin, allowing a local enteric antibacterial activity with minimal risk of systemic toxicity. Third, rifaximin acts as a gut-specific ligand for human Pregnane X receptor (PXR) [42]. PXR is a ligand-activated transcriptional factor, and nuclear receptor expressed universally along the gut-liver-axis [43]. Rifaximin downregulates the TLR4/NF-kB pathway by inducing endotoxin through a PXR-dependent manner [44] and finally improves the hepatic inflammation. Therefore, rifaximin-activated PXR may recover the intestinal barrier function by reducing intestinal permeability and gut endotoxin leakage in patients with cirrhosis. A recent clinical report also showed that rifaximin has a role in gut barrier repair by ameliorating bacterial translocation and systemic endotoxemia in patients with cirrhosis [45], which may improve the hepatic reserve.

In this study, rifaximin was effective and useful for patients with cirrhosis (Child–Pugh C), suggesting that rifaximin is equally effective regardless of the hepatic reserve. No study has reported the hepatic reserve of rifaximin to date. The result of this study is important that the long-term rifaximin treatment is found effective and safe.

One of the limitations of this study was that neuropsychological examination was not performed to diagnose covert HE, but ammonia levels were used, and the improvement of covert HE cannot be proven objectively. Moreover, the criteria for administering rifaximin differ by the attending physician and are not standardized. Previously reported studies of the Japanese population comprised a relatively small number of patients at a single center [15]; however, this study enrolled a large number of patients from multiple centers. This study excluded patients who discontinued rifaximin until 12 months; therefore, the exact incidence of AEs due to rifaximin remains known. Therefore, the prospective effect of rifaximin should be examined in the near future.

In the current study, liver function, renal function, and electrolytes were evaluated to assess the effects of rifaximin. AST, ALT, T-BIL, BUN, Cre, eGFR, Na, K, and MELD scores remained unchanged, whereas ALB, PT%, and ALBI scores statistically increased after 3 or more months as compared with the pretreatment, suggesting that the long-term rifaximin administration did not affect renal function and electrolytes and improve the hepatic reserve. In patients with Child–Pugh C, rifaximin administration did not affect renal function and electrolytes in the long term and improved the hepatic reserve at 3 months.

## 5. Conclusions

Long-term rifaximin treatment showed sustained efficacy for patients with HE and resulted in no severe AEs. Moreover, rifaximin improved the hepatic reserve.

## Figures and Tables

**Figure 1 jcm-11-01571-f001:**
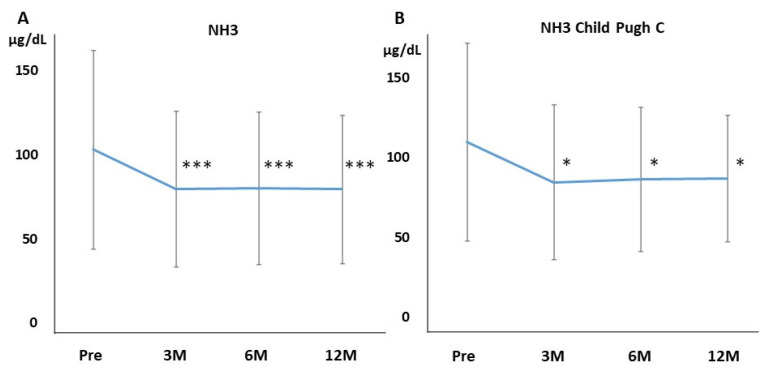
Changes in serum ammonia levels after rifaximin administration. (**A**) The ammonia levels significantly improved following rifaximin therapy from 102.9 ± 59.0 μg/dL to 79.7 ± 46.3 μg/dL, 80.1 ± 45.3 μg/dL, and 79.3 ± 44.1 μg/dL after 3, 6, and 12 months, respectively. (**B**) The ammonia levels of patients with Child–Pugh C significantly improved following rifaximin therapy from 109.7 ± 61.9 μg/dL to 84.5 ± 48.6 μg/dL, 86.5 ± 45.4 μg/dL, and 86.9 ± 39.5 μg/dL after 3, 6, and 12 months, respectively. A statistically significant difference was observed between the values at baseline and at 3, 6, 12 months. NH3, ammonia; Pre, before rifaximin treatment; 3M, after 3 months of rifaximin treatment; 6M, after 6 months of rifaximin treatment; 12M, after 12 months of rifaximin treatment. * *p* < 0.05, *** *p* < 0.001.

**Figure 2 jcm-11-01571-f002:**
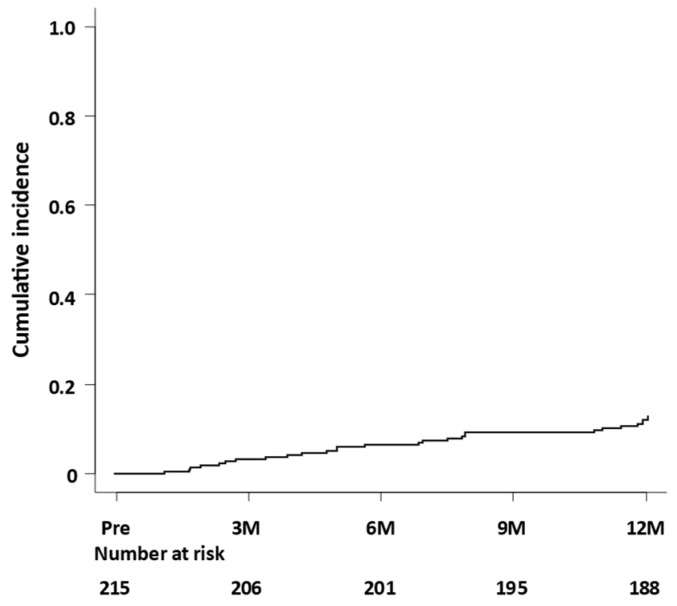
Cumulative incidence of the occurrence of hepatic encephalopathy. Pre, before rifaximin treatment; 3M, after 3 months of rifaximin treatment; 6M, after 6 months of rifaximin treatment; 9M, after 9 months of rifaximin treatment; 12M, after 12 months of rifaximin treatment.

**Figure 3 jcm-11-01571-f003:**
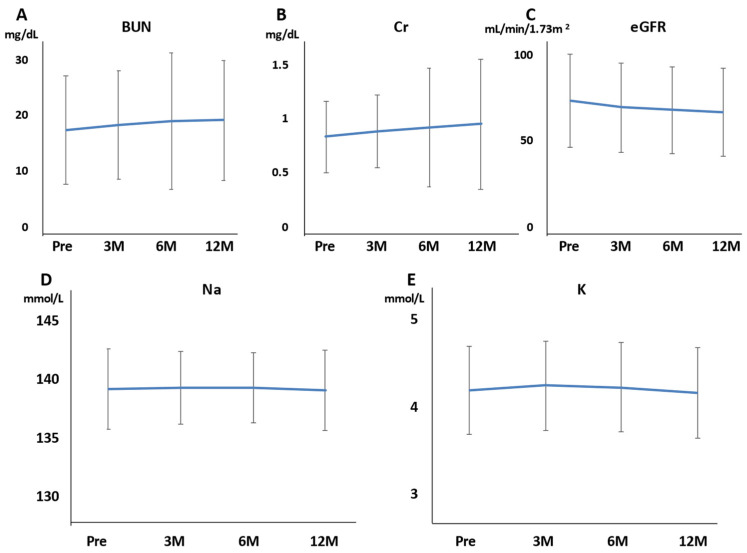
Effects of rifaximin on renal function and electrolytes in 215 patients with hepatic encephalopathy. (**A**) BUN; (**B**) Cr; (**C**) eGFR; (**D**) Na; (**E**) K. BUN, Cr, eGFR, Na, and K did not differ after rifaximin administration. BUN, blood urea nitrogen; Cr, creatinine; eGFR, estimated glomerular filtration rate; Na, sodium; K, potassium; Pre, before rifaximin treatment; 3M, after 3 months of rifaximin treatment; 6M, after 6 months of rifaximin treatment; 12M, after 12 months of rifaximin treatment.

**Figure 4 jcm-11-01571-f004:**
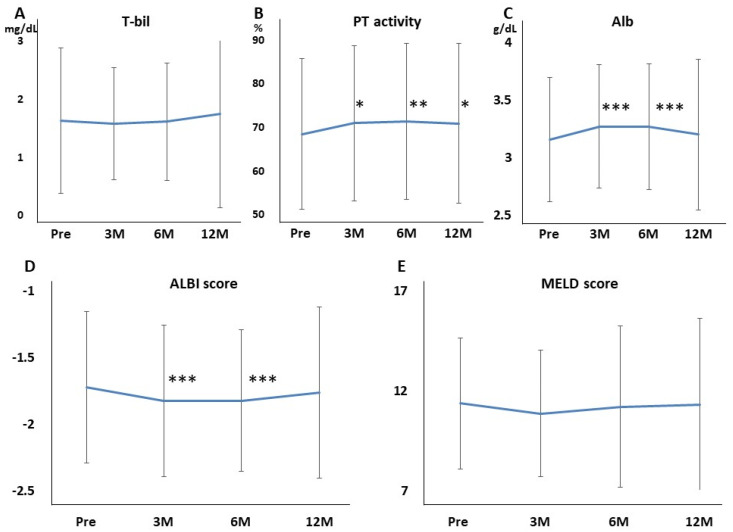
Effects of rifaximin on the hepatic reserve in 215 patients with hepatic encephalopathy. (**A**) T−bil; (**B**) PT activity; (**C**) Alb; (**D**) ALBI score; (**E**) MELD score. The PT activity, Alb level, and ALBI score improved after rifaximin administration. T−bil, total bilirubin; PT, prothrombin; Alb, albumin; ALBI, albumin−bilirubin; MELD, model for end−stage liver disease; Pre, before rifaximin treatment; 3M, after 3 months of rifaximin treatment; 6M, after 6 months of rifaximin treatment; 12M, after 12 months of rifaximin treatment. * *p* < 0.05 compared with before rifaximin treatment, ** *p* < 0.01 compared with before rifaximin treatment, *** *p* < 0.001 compared with before rifaximin treatment.

**Figure 5 jcm-11-01571-f005:**
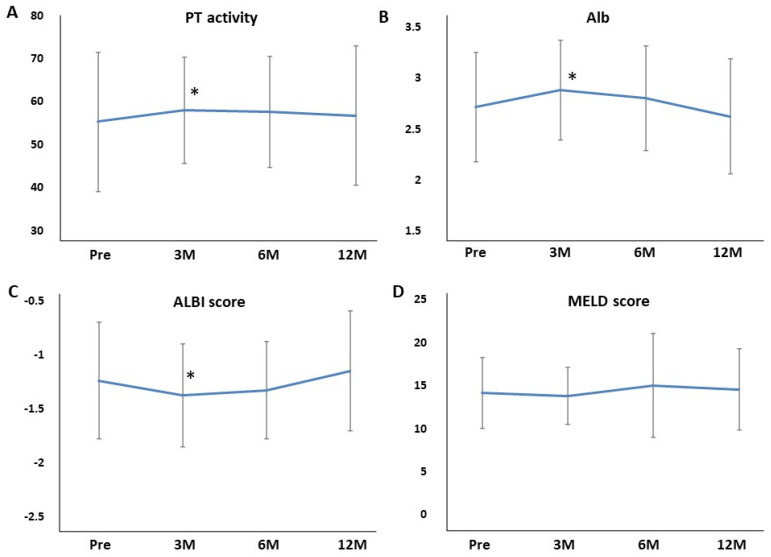
Effects of rifaximin on the hepatic reserve in 27 patients with hepatic encephalopathy limited to Child−Pugh C. (**A**) PT activity; (**B**) Alb; (**C**) ALBI score; (**D**) MELD score. The PT activity, Alb level, and ALBI score improved after 3 months of rifaximin administration. PT, prothrombin; Alb, albumin; ALBI, albumin–bilirubin; MELD, model for end−stage liver disease; Pre, before rifaximin treatment; 3M, after 3 months of rifaximin treatment; 6M, after 6 months of rifaximin treatment; 12M, after 12 months of rifaximin treatment. * *p* < 0.05 compared with before rifaximin treatment.

**Table 1 jcm-11-01571-t001:** Clinical characteristics of the patients.

Variables	(*n* = 215)
Age (years)	69.7 ± 11.6
Sex (male/female)	138/77
Body mass index (kg/m^2^)	24.8 ± 4.7
Etiology (HBV/HCV/Alcohol/NASH/others)	22/37/66/29/61
Child–Pugh (A/B/C)	14/174/27
History of hepatic encephalopathy (yes/no)	163/52
Presence of overt hepatic encephalopathy (yes/no)	34/181
Administration of lactulose (yes/no)	116/99
Presence of esophagogastric varices (yes/no)	119/96
Presence of a portosystemic shunt (yes/no/unknown)	107/106/2
Presence of splenomegaly (yes/no/splenectomy)	160/54/1
Presence of ascites (yes/no)	79/136
Presence of spontaneous bacterial peritonitis (yes/no)	0/215
Presence of hepatocellular carcinoma (yes/no)	74/141
Stage of hepatocellular carcinoma (1/2/3/4)	27/30/11/6

HBV, hepatitis B virus; HCV, hepatitis C virus; NASH, nonalcoholic steatohepatitis.

**Table 2 jcm-11-01571-t002:** Laboratory examinations at pretreatment.

Variables	(*n* = 215)
Hemoglobin (g/dL)	11.8 ± 2.1
Platelet (×10^4^/μL)	10.9 ± 5.7
Prothrombin activity (%)	68.5 ± 17.3
Serum albumin (g/dL)	3.2 ± 0.5
Aspartate aminotransferase (U/L)	46.7 ± 24.9
Alanine aminotransferase (U/L)	29.7 ± 15.9
Total bilirubin (mg/dL)	1.6 ± 1.2
Serum ammonia (μg/dL)	102.9 ± 59.1
Blood urine nitrogen (mg/dL)	17.3 ± 9.7
Serum creatine (mg/dL)	0.8 ± 0.3
Estimated glomerular filtration (mL/min/1.73 m^2^)	72.8 ± 27.0
Serum sodium (mEq/L)	139.2 ± 3.4
Serum potassium (mEq/L)	4.2 ± 0.5
ALBI score	−1.72 ± 0.57
MELD score	11.0 ± 3.3

ALBI, albumin–bilirubin; MELD, model for end-stage liver disease.

**Table 3 jcm-11-01571-t003:** Prediction of the occurrence of hepatic encephalopathy.

Variables	Univariate		Multivariate	
	HR (95% CI)	*p*-Value	HR (95% CI)	*p*-Value
Age (per year increase)	1.00 (0.96–1.03)	0.909		
Sex (male)	0.52 (0.24–1.12)	0.094	0.61 (0.27–1.37)	0.234
Etiology (HCV vs. except HCV)	0.71 (0.28–1.78)	0.464		
Child–Pugh C (vs. A, B)	1.66 (0.63–4.41)	0.307		
Presence of esophagogastric varices	1.28 (0.58–2.82)	0.542		
Presence of a portosystemic shunt	2.42 (1.05–5.58)	0.037	1.68 (0.68–4.13)	0.262
Presence of splenomegaly	1.02 (0.41–2.56)	0.965		
Presence of ascites	1.21 (0.53–2.76)	0.645		
Presence of HCC	2.32 (0.87–6.15)	0.091	1.92 (0.69–5.32)	0.209
Stage of HCC	1.13 (0.49–2.94)	0.724		
Alanine aminotransferase at baseline	1.00 (0.97–1.02)	0.735		
Blood urine nitrogen at baseline	1.01 (0.98–1.05)	0.548		
Creatinine at baseline	1.33 (0.47–3.78)	0.592		
Serum sodium at baseline	1.06 (0.94–1.20)	0.323		
Serum albumin at baseline	0.44 (0.21–0.90)	0.025	0.58 (0.23–1.44)	0.241
Total bilirubin at baseline	1.14 (0.88–1.49)	0.316		
Prothrombin activity at baseline	0.96 (0.95–0.99)	<0.001	0.98 (0.95–1.00)	0.049

HR, Hazard Ratio; CI, confidence interval; HCV, hepatitis C virus; HCC, hepatocellular carcinoma.

## Data Availability

Not applicable.

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
