# Peer review of "Long-Term Efficacy and Safety of Rifaximin in Japanese Patients with Hepatic Encephalopathy: A Multicenter Retrospective Study"

_jcm, 2022, doi:10.3390/jcm11061571_

Round 1

Reviewer 1 Report

Long-term efficacy and safety of rifaximin in Japanese patients with hepatic encephalopathy: A multicenter retrospective study, by Kawaratani, et al.

They evaluated the long-term efficacy and safety of rifaximin in patients with LC. Although several data of long-term rifaximin treatment are currently available in Western countries, their results are highly informative in Japanese hepatologist, which can be acceptable in JCM.

Please address the following points.

#1. Table 1 and 2.

In BMI, units should be provided.

In AST, ALT and eGFR, units should be provided.

#2. Figure 2-4.

Error bar of T-BIL at 12 months should be provided.

The vertical axis of each parameter should contain the units.

#3. Table 3.

  1. Kaplan-Meier analysis for the HE occurrence should be applied in the univariate analysis. Time interval between pre and the occurrence date of HE should differ in each patient. In addition, Cox proportional hazard model should be used in the multivariate analysis because occurrence of HE is time-dependent. The authors should re-analyze accordingly.
  2. Factors with a P value <0.1 were entered into the multivariate analysis. In the statistical section, the authors should describe this point.

#4. References

Please consider to quote the articles below.

Hepatol Res. 2021 Jul;51(7):725-749. 

Author Response

  1. Table 1 and 2. In BMI, units should be provided. In AST, ALT and eGFR, units should be provided.

→Thank you for your comment. In accordance with reviewer’s comment, we added the exact units for BMI, AST, ALT, and eGFR in Table 1 and 2.

  1. Figure 2-4. Error bar of T-BIL at 12 months should be provided. The vertical axis of each parameter should contain the units.

→ Thank you for your comments. In accordance with reviewer’s comment, we corrected Figure 2-4.

  1. Table 3.
  1. Kaplan-Meier analysis for the HE occurrence should be applied in the univariate analysis. Time interval between pre and the occurrence date of HE should differ in each patient. In addition, Cox proportional hazard model should be used in the multivariate analysis because occurrence of HE is time-dependent. The authors should re-analyze accordingly.
  2. Factors with a P value <0.1 were entered into the multivariate analysis. In the statistical section, the authors should describe this point.

 →1. Thank you for your important comment. As the reviewer pointed out, we re-analysed by using Cox proportional hazard model in the univariate and multivariate analysis.  The occurrence of HE were evaluated by using Kaplan-Meier curve.

→2. And we added the sentence that the factors associated with the occurrence of HE, a P value <0.1 were entered into the multivariate analysis, in the statistical section.

  1. References. Please consider to quote the articles below.

Hepatol Res. 2021 Jul;51(7):725-749.

→Thank you for your comment. In accordance with the reviewer’s comment, we added the reference in the revised manuscript.

Reviewer 2 Report

The authors have studied the long-term efficacy and safety of rifaximin in Japanese patients with hepatic encephalopathy (HE). The main limitation of study is its novelty. PMID: 31293721, a single center study in Japanese population has demonstrated the safety and efficacy of rifaximin in HE for a period of six months. Moreover  similar studies has been demonstrated in other population. Although authors have acknowledged the earlier, the findings derived from this stud does not substantiate or enhance the existing knowledge. 

1) The manuscript must be thoroughly proof-read. For instances line 46-51, line 122-124 should be deleted, line 66-68, line 77-78,line 102-104 should be checked for sentence correctness.  The first paragraph of discussion appears to be repeated. Line 179 the sentence should appear below figure legends.  in discussion the references or citations are missing parenthesis. line 191 shows 28 patients of child pugh c, however table 1 shows 27 patients with child pugh C. Such typographical errors are found throughout the manuscript. Most of abbreviations has be to expanded at first instances 

2) For logistic regression analysis no cut-off values are considered for the variables. The data should be reanalyzed. In addition how many patients had occurrence of HE is not mentioned in the result section.  In the abstract, the number of overt HE incidence is reported as 15 cases and hospitalization as 7 cases. Similarly in discussion it is reported as 15cases.  So, during logistic regression analysis how many cases were considered ?. is not clear. 

3) How many times a day rifaximin was prescribed in  the study population. 

4) One of the key data would be the assessment of liver enzymes over a different time period which is lacking in this study, which would have strengthen the study. 

Author Response

Major points

  • The manuscript must be thoroughly proof-read. For instances line 46-51, line 122-124 should be deleted, line 66-68, line 77-78, line 102-104 should be checked for sentence correctness.
  1. The first paragraph of discussion appears to be repeated. Line 179 the sentence should appear below figure legends.
  2. In discussion the references or citations are missing parenthesis.
  3. line 191 shows 28 patients of child pugh c, however table 1 shows 27 patients with child pugh C.
  4. Such typographical errors are found throughout the manuscript. Most of abbreviations has be to expanded at first instances

→Thank you for your careful comments. In accordance with reviewer’s comment, we changed the sentences in the revised manuscript. 1. We changed the sentences and performed English language editing. 2. We changed the discussion session in the revised manuscript. 3. We added the parenthesis. 4. It was our mistyping, 27 patients is the correct. 5. We changed the sentences and performed English language editing.

  1. For logistic regression analysis no cut-off values are considered for the variables. The data should be reanalyzed. In addition, how many patients had occurrence of HE is not mentioned in the result section.  In the abstract, the number of overt HE incidence is reported as 15 cases and hospitalization as 7 cases. Similarly in discussion it is reported as 15cases.  So, during logistic regression analysis how many cases were considered? is not clear.

→Thank you for your comment. The occurrence of HE was evaluated by using Kaplan-Meier curve. And the numbers at risk of HE occurrence are shown in Table 2. And, we re-analysed by using Cox proportional hazard model in the univariate and multivariate analysis.  

  1. How many times a day rifaximin was prescribed in the study population.

 →Thank you for your comment. 6 tablets of 200mg rifaximin were administered trice dairy. We added the sentence in revised manuscript.

  1. One of the key data would be the assessment of liver enzymes over a different time period which is lacking in this study, which would have strengthen the study

→ Thank you for your valuable comment. We analysed the time course of liver enzyme (AST and ALT), however, there were no difference in time coarse. We wrote the sentence in the result session.

Round 2

Reviewer 1 Report

The revised article seems to be nicely revised.

Reviewer 2 Report

The authors have taken effort to significantly imporve the manuscript and made made the revsions in the data analysis.